# Measurement and In-Depth Analysis of Higher Harmonic Generation in Aluminum Alloys with Consideration of Source Nonlinearity

**DOI:** 10.3390/ma16124453

**Published:** 2023-06-18

**Authors:** Hyunjo Jeong, Hyojeong Shin, Shuzeng Zhang, Xiongbing Li

**Affiliations:** 1Department of Mechanical Engineering, Wonkwang University, Iksan 54538, Republic of Korea; 2Graduate School of Flexible and Printable Electronics, Jeonbuk National University, Jeonju 54896, Republic of Korea; hjshin95@jbnu.ac.kr; 3School of Traffic and Transportation Engineering, Central South University, Changsha 410083, China; sz_zhang@csu.edu.cn (S.Z.); lixb213@csu.edu.cn (X.L.)

**Keywords:** nonlinear ultrasound, higher harmonic generation, source nonlinearity, cubic nonlinearity, quadratic nonlinearity

## Abstract

Harmonic generation measurement is recognized as a promising tool for inspecting material state or micro-damage and is an ongoing research topic. Second harmonic generation is most frequently employed and provides the quadratic nonlinearity parameter (β) that is calculated by the measurement of fundamental and second harmonic amplitudes. The cubic nonlinearity parameter ($\beta^{2}$), which dominates the third harmonic amplitude and is obtained by third harmonic generation, is often used as a more sensitive parameter in many applications. This paper presents a detailed procedure for determining the correct $\beta^{2}$ of ductile polycrystalline metal samples such as aluminum alloys when there exists source nonlinearity. The procedure includes receiver calibration, diffraction, and attenuation correction and, more importantly, source nonlinearity correction for third harmonic amplitudes. The effect of these corrections on the measurement of$\;\beta^{2}$ is presented for aluminum specimens of various thicknesses at various input power levels. By correcting the source nonlinearity of the third harmonic and further verifying the approximate relationship between the cubic nonlinearity parameter and the square of the quadratic nonlinearity parameter (β∗β), $\beta^{2}\approx \beta *\beta$, the cubic nonlinearity parameters could be accurately determined even with thinner samples and lower input voltages.

## 1. Introduction

The principle of nonlinear ultrasonic inspection is to generate and detect higher harmonic waves having integer multiples of the incident fundamental wave frequency when a fundamental wave of finite amplitude is injected into the transmitter and propagates in a test object. Since the degree of generated nonlinear wave components depends on the material state, the magnitude and characteristics of the received second or third harmonic signal can be correlated with microscopic damage or defects in testing components. The second harmonic, which has a frequency twice the fundamental frequency, has been mostly used so far for nonlinear ultrasound examinations. The quadratic nonlinearity parameter (β), calculated by the fundamental and second harmonic wave amplitudes, is now widely accepted as a quantitative index of damage.

The generation and use of the third harmonic that has a frequency three times the fundamental frequency are also possible, and it is known that the sensitivity to damage or defects is much higher than that of the second harmonic. The cubic nonlinearity parameter ($\beta^{2}$) is often used as a more sensitive and qualitative parameter by generating third harmonic waves. Actually, plane wave solutions for the nonlinear wave equation provide an approximate relationship, $\beta^{2}\approx \beta *\beta$, between the cubic nonlinearity parameter ($\beta^{2}$) and the square of the quadratic nonlinearity parameter (β∗β) [1]. Compared to the second harmonic wave or β, the $\beta^{2}\approx \beta *\beta$ relation and the high frequency nature of the third harmonic wave are expected to provide a much higher sensitivity and resolution to the same defect or damage. The advantages of using the cubic nonlinearity parameter for fatigue cracks [2,3], plastic deformation [4,5,6], microstructures [7,8,9], dislocation [10], and precipitation [11,12] of metals were discussed by comparing the relevant results of the quadratic nonlinearity parameter. In addition to longitudinal waves, third harmonic generation using Lamb waves and Rayleigh surface waves was studied for different application purposes. Lissenden et al. [13] studied the effect of microstructure evolution on the higher harmonic generation of guided waves. Zhao et al. [14] used third harmonic Lamb waves for early fatigue damage detection. Li et al. [15] and Wen et al. [16] applied third harmonic shear horizontal waves for material degradation monitoring. Achenbach and Wang [17] and Wang and Achenbach [18] used third harmonic surface waves for the characterization of incompressible material of cubic nonlinearity and interface conditions, respectively.

Measuring the cubic nonlinearity parameter using third harmonic wave amplitude is more challenging in many ways. One major problem is the generation of source nonlinearity. When a piezoelectric element with a fundamental resonance frequency *f* is used as a transmitter, it also resonates at the 3rd, 5th, 7th, … overtones. Therefore, a large frequency component of 3*f* is also generated and propagates as an additional fundamental wave of 3*f* when a toneburst pulse of frequency *f* excites the transmitter. This is called “source nonlinearity”, and when a finite amplitude signal is applied as an input source, it is superimposed with the damage- or defect-induced third harmonic wave. The amplitude of this source nonlinearity can be many times greater than that of the nonlinear third harmonic; therefore, it is very important to carefully check for its presence and remove it appropriately for an accurate assessment of the material state or $\beta^{2}$.

The absolute measurement of β generally includes receiver calibration, sound beam diffraction correction, and material attenuation correction. The absolute measurement of β is now well established [19,20], but to the best of our knowledge, the absolute $\beta^{2}$ has never been measured before for solid samples, especially with consideration of source nonlinearity. Thompson theoretically obtained the relationship between β and $\beta^{2}$ for various materials [1]. In the case of brittle fused quartz, it was shown that the relationship of $\beta^{2}\approx \beta *\beta$ is almost satisfied, and some other materials also approximately satisfy this relationship. He also experimentally verified this relationship for brittle materials but could not draw any conclusions on ductile materials such as aluminum. Recently, Jeong et al. [21] demonstrated that this approximate relationship holds in water through simultaneous measurement of the absolute nonlinearity parameters.

The purpose of this paper is to accurately measure the cubic nonlinearity parameter ($\beta^{2}$) and compare it with the square of the quadratic nonlinearity parameter (β∗β) by considering the source nonlinearity associated with the generation of second and third harmonic waves in ductile polycrystalline materials such as aluminum alloys. To this end, the inclusion of source nonlinearity in the measured amplitude of higher harmonics is confirmed and is appropriately corrected in the calculation of the nonlinearity parameters. The effect of source nonlinearity correction on $\beta^{2}$ is then analyzed through aluminum specimens of various thicknesses at various input power levels.

The rest of this paper is organized as follows. Section 2 briefly introduces the plane wave solutions for the nonlinear wave equation and the definition of quadratic (β) and cubic ($\beta^{2}$) nonlinearity parameters with corrections for attenuation and diffraction. Section 3 describes the elements of absolute measurement of nonlinearity parameters: Receiver calibration measurement, harmonic generation measurement, absolute displacement measurement, diffraction, and attenuation corrections, and check for source nonlinearity and correction. Section 4 presents the in-depth analysis of the experimental results with an emphasis on the correction of source nonlinearity for accurate comparison of $\beta^{2}\;$and β∗β. Lastly, conclusions are outlined in Section 5.

## 2. Plane Wave Solutions and Nonlinearity Parameters

The normal stress can be expressed in terms of strain/displacement in the *x*-direction for pure longitudinal wave propagation in an isotropic solid with cubic nonlinearity [18]
(1)$$\sigma_{x}=\rho c^{2}\left[\frac{\partial u}{\partial x}-\frac{\beta }{2}{\left(\frac{\partial u}{\partial x}\right)}^{2}-\frac{\gamma }{3}{\left(\frac{\partial u}{\partial x}\right)}^{3}\right]$$
where ρ is the density, *c* is the wave speed, and β and γ are nonlinearity parameters. β and γ are given by β = −(3 + *C*_111_/*C*_11_) and γ = −(3/2 + 3*C*_111_/*C*_11_+*C*_1111_/2*C*_11_), where *C*_11_, *C*_111_, and *C*_1111_ are the second-, third-, and fourth-order elastic constants.

Substituting Equation (1) into the equation of motion leads to the displacement equation of motion governing the longitudinal wave propagation [1],
(2)$$\frac{1}{c^{2}}\frac{\partial^{2}u}{\partial t^{2}}-\frac{\partial^{2}u}{\partial x^{2}}=-\left[\beta \left(\frac{\partial u}{\partial x}\right)+\gamma {\left(\frac{\partial u}{\partial x}\right)}^{2}\right]\frac{\partial^{2}u}{\partial x^{2}}$$

The perturbation method can be used to obtain the solutions of Equation (2), where the total solution is expressed as $u=u_{1}+u_{2}+u_{3}$ with the assumption of $u_{1}\gg u_{2}\gg u_{3}$. Here, $u_{1},\;u_{2},\;\mathrm{and}\;u_{3}$ are the displacement solutions for the fundamental, second, and third harmonic waves, respectively. The governing equations for the first three waves are obtained as
(3)$$\frac{1}{c^{2}}\frac{\partial^{2}u_{1}}{\partial t^{2}}-\frac{\partial^{2}u_{1}}{\partial x^{2}}=0$$
(4)$$\frac{1}{c^{2}}\frac{\partial^{2}u_{2}}{\partial t^{2}}-\frac{\partial^{2}u_{2}}{\partial x^{2}}=-\beta \frac{\partial u_{1}}{\partial x}\frac{\partial^{2}u_{1}}{\partial x^{2}}$$
(5)$$\frac{1}{c^{2}}\frac{\partial^{2}u_{3}}{\partial t^{2}}-\frac{\partial^{2}u_{3}}{\partial x^{2}}=-\beta \left[\frac{\partial u_{1}}{\partial x}\frac{\partial^{2}u_{2}}{\partial x^{2}}+\frac{\partial u_{2}}{\partial x}\frac{\partial^{2}u_{1}}{\partial x^{2}}\right]-\gamma \frac{\partial^{2}u_{1}}{\partial x^{2}}{\left[\frac{\partial u_{1}}{\partial x}\right]}^{2}$$

Equation (1) is the traditional linear wave equation for the fundamental wave $u_{1}\;$of which the solution is the plane wave with initial source amplitude $U_{0}$, wave number *k*, and angular frequency ω:(6)$$u_{1}=U_{0}\sin\left(kx-\omega t\right)$$

Substituting $u_{1}$ into Equations (4) and (5) and performing some algebra give the solutions for $u_{2}\;$and $u_{3}$ [18,22]:(7)$$u_{2}=\frac{\beta U_{0}^{2}k^{2}x}{8}\cos2\left(kx-\omega t\right)$$
(8)$$u_{3}=\frac{\beta^{2}U_{0}^{3}k^{4}x^{2}}{32}\sin3\left(kx-\omega t\right)$$

The fundamental wave amplitude is *U*_1_ = *U*_0_ in Equation (6), and the second harmonic wave amplitude is given by $U_{2}=\frac{\beta U_{0}^{2}k^{2}x}{8}$ from Equation (7). The third harmonic wave amplitude generally depends on both β and γ [22]. It should be noted that the third harmonic amplitude $U_{3}\approx \frac{\beta^{2}U_{0}^{3}k^{4}x^{2}}{32}$ in Equation (8) is valid when (*kx*) is large. However, $U_{3}\approx \frac{\gamma U_{0}^{3}k^{3}x}{32}$ in the cases where γ is the large and dominant factor, which directly links the third harmonic amplitude to γ [22].

It is worth noting that, if γ has a major effect on the third harmonic amplitude, the following conditions must be satisfied: $\gamma \gg \beta^{2}$ and kx≪1. However, in most nonlinear ultrasound measurements using a finite amplitude method, the frequency in megahertz is usually used, and the third harmonic amplitude is very small at a short propagation distance; therefore, kx≫1 should be employed. A reliable generation of the third harmonic amplitude actually requires kx≫500. Therefore, third harmonic generation due to γ cannot be measured experimentally under the existing nonlinear experimental conditions, and this is why the third harmonic wave amplitude in Equation (8) is expressed in terms containing only $\beta^{2}$ [21,22]. In this study, we call $\beta^{2}$ the cubic nonlinearity parameter because it actually dominates third harmonic generation.

The actual displacement of a wave generated and received by finite-size transducers can be expressed by the plane wave amplitude modified by diffraction and attenuation effects [21]
(9)$$U_{1}\left(x\right)=U_{0}\left[D_{1}\left(x\right)\right]\left[M_{1}\left(x\right)\right]$$
(10)$$U_{2}\left(x\right)=\frac{\beta U_{0}^{2}k^{2}x}{8}\left[D_{2}\left(x\right)\right]\left[M_{2}\left(x\right)\right]\;$$
(11)$$U_{3}\left(x\right)=\frac{\beta^{2}U_{0}^{3}k^{4}x^{2}}{32}\left[D_{3}\left(x\right)\right]\left[M_{3}\left(x\right)\right]$$
where $D_{i}\;\mathrm{and}\;M_{i},\;i=1,\;2,\;3$ represent the diffraction and attenuation corrections at a propagation distance *x*, respectively.

Equations (9)–(11) provide a practical means to determine the displacement-based nonlinearity parameter β in Equation (10) and $\beta^{2}$ in Equation (11). β can be determined by measuring the fundamental and second harmonic amplitudes using Equations (10) and (11). We call β the quadratic or second-order nonlinearity parameter. $\beta^{2}$ can be determined by measuring the fundamental and third harmonic amplitudes using Equations (9) and (12). We call $\beta^{2}$ the cubic or third-order nonlinearity parameter. The quadratic and cubic nonlinearity parameters at the propagation distance *x* are defined as [21]
(12)$$\beta \left(x\right)=\frac{8U_{2}\left(x\right)}{k^{2}xU_{1}^{2}\left(x\right)}\left[\frac{D_{1}^{2}\left(x\right)}{D_{2}\left(x\right)}\right]\left[\frac{M_{1}^{2}\left(x\right)}{M_{2}\left(x\right)}\right]\;$$
(13)$$\beta^{2}\left(x\right)=\gamma \left(x\right)=\frac{32U_{3}\left(x\right)}{k^{4}x^{2}U_{1}^{3}\left(x\right)}\left[\frac{D_{1}^{3}\left(x\right)}{D_{3}\left(x\right)}\right]\left[\frac{M_{1}^{3}\left(x\right)}{M_{3}\left(x\right)}\right]$$

If the diffraction and attenuation corrections are neglected, Equations (12) and (13) will be reduced to the definitions of nonlinearity parameters based on the pure plane wave solutions.

## 3. Elements of Absolute Nonlinearity Parameter Measurement

### 3.1. Receiver Calibration Measurement

In harmonic generation measurement, the output signal received from the receiving transducer is measured in the form of electrical voltage. However, the nonlinearity parameters in Equations (12) and (13) are defined in terms of displacement. The purpose of receiver calibration is to obtain the transfer function $H_{r}\left(\omega \right)$ that converts the current output$\;I_{out}\left(\omega \right)$ to the displacement output $U\left(\omega \right)$. There exist reciprocity-based calibration methods [23,24,25] where a piezoelectric transducer is mounted on the receive side of a sample in a pulse-echo configuration, and the voltage and current are measured at the input and output ports of the transducer. The calibration method used in this study is a simplified version of the existing method [23], which requires only the current measurements at the input and output ports of the receiving transducer. The diffraction and attenuation corrections should also be performed for accurate determination of $H_{r}\left(\omega \right)$. Receiver calibration is performed on each specimen, and the receiver pressurization remains the same during the entire test of that specimen. A detailed derivation of the receiver transfer function and the experimental procedure are described elsewhere [19].

### 3.2. Harmonic Generation Measurement

After the receiver calibration measurement is completed, a finite amplitude through a transmission test is conducted for harmonic generation measurement. The transmitting transducer (T) is a single-crystal lithium niobate (LiN) of 5 MHz center frequency and 9.5 mm diameter, while the receiving transducer (R) is a broadband commercial transducer of the same diameter. The two transducers are aligned coaxially through the solid sample for maximum output signal capture. A series of calibration measurements and harmonic generation measurements are performed on aluminum samples of various thicknesses.

The samples used in this study are commercially available aluminum alloy 6061-T6. Six different samples in thickness were prepared for the nonlinearity parameter measurements: 2, 4, 6, 8, 10, and 12 cm. Each piece was obtained by cutting in the shape of a rectangular parallelepiped from a large-sized circular bar. The size of the cross section is a square of 4 cm × 4 cm. The top and bottom surfaces, where the transducers are installed, were further processed to be flat and parallel to each other.

Figure 1 shows the block diagram of harmonic generation measurement. A high-power toneburst pulser (RPR-4000, RITEC, Warwick, RI, USA) is used to produce a high-voltage, 20-cycle toneburst tuned to the fundamental frequency (5 MHz) that is applied to the transmitter via a 50 Ohm high-power feedthrough and a high-power stepped attenuator. The receiver side is comprised of the receiving probe coupled to a 50 Ohm load via a current probe (Tektronix CT-2, Tektronix, Wilsonville, OR, USA) with the through-transmitted toneburst signal captured on a digital storage oscilloscope (WaveSurfer 3024, Teledyne LeCroy, Chestnut Ridge, NY, USA). The current probe used in this study provides 1 mV per 1 milliamp when terminated in 50 Ohm. Nine different input voltages (from 0 to 40 power levels in 5 level step) are applied from the high-power pulser. These input power levels correspond to approximately 30–300 $V_{\mathrm{peak}}\;$at the transmitter.

### 3.3. Measurement of Harmonic Displacement Amplitude

The receiving transducer is first calibrated using the simplified self-reciprocity technique [19], which can minimize the errors induced by impedance mismatch. The purpose of receiver calibration is to find the transfer function that converts the output current to the absolute displacement. The measured current signal in the subsequent harmonic generation experiment is convolved with the transfer function in the frequency domain, and the fundamental, second harmonic, and third harmonic components are separately inverse Fourier transformed to obtain the absolute displacement amplitude of each component in the time domain. The measured displacement amplitudes are then used to calculate the absolute nonlinearity parameter according to Equations (12) and (13). Experiments are performed to determine β and $\beta^{2}$ of each sample using the fundamental and higher harmonic displacement amplitudes extracted from the same output signal acquired at each input power level.

### 3.4. Diffraction and Attenuation Corrections

The original definition of the nonlinearity parameter is based on the plane wave displacement solutions for the one-dimensional nonlinear wave equation. In most nonlinear ultrasound experiments, however, the sound beam is generated by a finite-size transducer, and it is not purely a plane wave. Therefore, one needs to adjust the amplitudes of the actual acoustic fields to their plane wave values before they are used to determine the nonlinearity parameter. This is the effect referred to as the diffraction correction and introduced in the measured displacement amplitudes in Equations (9)–(11).

Diffraction correction is defined as the amplitude of the actual wave divided by that of the plane wave with both wave amplitudes received at the same propagation distance in a nonattenuating medium. Diffraction effects generally depend on the size of the transmitter and receiver, frequency, and propagation distance. An exact integral expression exists for the linear field when both transmitter and receiver sizes are the same [26,27]. The diffraction corrections for both fundamental and higher harmonic waves have been developed and can be efficiently used in a wide range of transmitter–receiver geometries [21]. The variation of diffraction correction as a function of propagation distance is shown in [21].

The amplitude of a wave propagating in a medium is also affected by attenuation, the loss of wave energy due to scattering and absorption, which generally depends on the frequency of the propagating wave. Since the measured wave amplitude deviates from that of a pure plane wave, the attenuation correction is also required in the measured displacement in Equations (9)–(11). The attenuation corrections for the fundamental and higher harmonic waves can be derived from the solutions of the one-dimensional Westervelt equation or Burger’s equation. They are explicitly given in [21] as a function of the attenuation coefficients and propagation distance.

### 3.5. Check of Source Nonlinearity and Correction

In a nonlinear ultrasonic measurement system, a low-frequency bandpass filter is frequently employed in the input stage to pass the fundamental wave and suppress the higher harmonic frequency components. In this study, this type of filter was not used to check the existence of the source nonlinearity related to the second and third harmonics.

It is known that a higher input voltage is required for the proper generation of the third harmonic amplitude in the test specimen along with sensitive broadband reception of the output signal [21]. An increase in the noise floor may occur as the input source level increases. This is another issue that can affect the measurement accuracy of the cubic nonlinearity parameter $\beta^{2}$. Therefore, it is important to check for the presence of source nonlinearity and make an appropriate correction.

In this study, we decided to use a lithium niobate (LiN) crystal instead of a transmitter in the form of a transducer in order to minimize source nonlinearity for the second harmonic and increase the efficiency of generating the third harmonic inside the specimen. In harmonic generation experiments, there are two main causes of source nonlinearity with respect to the third harmonic wave. First, it can be caused by the harmonics of the measurement system component such as a high-power amplifier. The generated harmonic will propagate as a fundamental wave with three times the fundamental frequency $3f_{0}.$ Second, when a finite amplitude narrowband toneburst with a fundamental frequency *f*_0_ is applied to the transmitter, a noticeable 3*f*_0_ component can be generated due to the odd harmonic resonance of the bare crystal at or close to 3*f*_0_. This also propagates as the linear wave of frequency 3*f*_0_. These two waves will be added up to the nonlinear third harmonic wave generated in the specimen, sometimes giving an excessively large value of third harmonic amplitude or the resulting cubic nonlinearity parameter.

Recently, phononic crystals or metamaterial surfaces have been proposed as frequency-filtering devices that could significantly decrease or eliminate unwanted harmonic waves by designing their bandgap structures at the desired frequency [28,29,30,31]. When such devices are inserted between the transmitting transducer and the specimen of interest, the designed bandgap should allow the fundamental frequency wave to propagate the specimen while inhibiting propagation at the second and/or third harmonic frequency before the incident wave enters the specimen.

In this study, the existence of source nonlinearity will be identified through the analysis of measured output signals, and the accuracy of the measured nonlinearity parameter will be improved through appropriate corrections of source nonlinearity. The detailed procedure for this correction and their effects on the nonlinearity parameter determination are discussed in the next section.

### 3.6. Comments on Contact Method of Nonlinear Ultrasound Testing

In contact nonlinear ultrasound testing, the contact and interface conditions between the specimen and the transducer can have a significant impact on the measured harmonic amplitudes. The contact and interface conditions may include the surface roughness of the specimen [28], the type/amount/contact holding time of the couplant [29], and the intensity of the contact pressure. In the case of transmission measurement, it is necessary to consider both the transmission side and the reception side. When measuring absolute nonlinearity parameters, it is necessary to ensure that the calibration measurement state of the receiver remains the same during the harmonic generation measurements. The first author of this paper conducted research on nonlinear ultrasound tests for many years and established a nonlinearity parameter measurement technology with excellent accuracy and repeatability. Some of them were used in this study as described below.

In order to minimize the effect of surface roughness, it is necessary to maintain the same surface roughness on each specimen as much as possible. The prepared specimens were machined so that the upper and lower surfaces were parallel. The surface roughness of each specimen was maintained at the same level as possible using a metal abrasive. A thin layer of couplant is applied to the transducer surface, and the transducer is maintained in a pressurized state with a constant pressure using a pressurization device. Experimental data are acquired after the pressed couplant reaches a steady state. This time usually takes several minutes. A specially designed pressurization fixture is used so that the receiver and transmitter are pressed separately. The pressurization state of the receiver during the calibration measurement remains the same throughout the harmonic generation measurement. A pin spring-type fixing and pressurization device is devised and used for the pressurization of the piezoelectric elements such as the bare crystals in the transmission side. Using this set of contact and boundary conditions, the quadratic nonlinearity parameter can be measured with less than 5% uncertainty, while the cubic nonlinearity parameter can be measured with less than 10% uncertainty.

## 4. Results and Discussion

### 4.1. Diffraction Correction and Attenuation Correction

Figure 2 shows the variation of the diffraction correction as a function of the propagation distance $D_{i},\;i=1,\;2,\;3$, which was calculated from Equations (24)–(26) of Ref [21]. The acoustic parameters used in the calculation are given in the figure caption. The effect of diffraction correction on nonlinearity parameter determination is investigated later.

Referring to Equations (21)–(23) in Ref [21], making attenuation corrections requires the information on the attenuation coefficients $\alpha_{1},\;\alpha_{2},\;and\;\alpha_{3}$ at the fundamental, second, and third harmonic frequencies, respectively. In the previous study [19], the attenuation coefficients $\alpha_{1}=4.6\;\mathrm{Np}/m$ and $\alpha_{2}=13.8\;\mathrm{Np}/m\;$for Al 6061 were extracted by applying a nonlinear least squares data fitting method without independent measurements of these coefficients. It was found that the frequency-dependent attenuation holds in the form of $\alpha_{f}=\alpha_{0}f^{m}$ with $\alpha_{0}=0.36$ and *m* = 1.585. This power law frequency-dependent attenuation provides $\alpha_{3}=26.24\;\mathrm{Np}/m$ at the third harmonic frequency. Figure 3 shows the variation of three attenuation corrections $M_{i},\;i=1,\;2,\;3\;$as a function of propagation distance.

### 4.2. Receiver Transfer Function

The receiver transfer function of each specimen was measured in the broadband pulse-echo testing configuration using the simplified calibration procedure described previously. Figure 4 shows the receiver transfer functions obtained for all six specimens after the diffraction and attenuation corrections were performed. The magnitude spectrum of the receiver transfer function $\left|H_{r}\left(\omega \right)\right|$ shows a bandwidth broad enough to cover the fundamental and higher harmonic frequencies from 5 to 18 MHz. Since the magnitude spectrum is also given as a function of the plane wave term $\exp\left(ikz\right)$, where *z* is the propagation distance, it shows a dependence on the sample thickness or propagation distance. At a given frequency, the spectral values become smaller as the sample thickness increases. Each transfer function will be used to convert the electrical output signal of the harmonic generation measurement into the absolute displacement signal from which displacement amplitudes are found to calculate the nonlinearity parameters.

### 4.3. Received Waveform and Frequency Spectrum

Right after the receiver calibration measurement, the harmonic generation measurement was performed using the finite amplitude through the transmission method. The purpose of this experiment was to obtain the displacement amplitudes of the fundamental and higher harmonic waves from which the nonlinearity parameters β and $\beta^{2}$ of each sample were determined. Figure 5a,b shows typical examples of the current output signal and its Fourier spectrum acquired from the 8 cm thick sample. In addition to the fundamental component at *f* = 5 MHz, the second and third harmonic components are clearly observed at 2*f* = 10 MHz and 3*f* = 15 MHz, respectively.

### 4.4. Calculation of Absolute Displacement

The frequency components of the output current, Figure 5b, is obtained by Fourier transforming the measured current signal, Figure 5a, and then convolved with the receiver transfer function $H_{r}\left(\omega \right)$ to calculate the frequency domain displacement spectrum. To extract the first three harmonic displacement components, three rectangular windows are used, and each windowed spectrum is inverse Fourier transformed to obtain the time domain displacement signal. A rectangular window with the frequency range of 4–6 MHz was used for the fundamental wave, while the frequency ranges of 9–11 MHz and 14–16 MHz were used for the second and third harmonic waves.

Figure 6 shows the extracted displacement waveforms for the first three harmonic waves. The average peak-to-pick displacement amplitudes $U_{1},\;U_{2},\;\mathrm{and}\;U_{3}$ are acquired from each of these figures and used in the subsequent calculation of $\beta \;\mathrm{and}\;\beta^{2}$.

### 4.5. Effects of Diffraction and Attenuation Corrections on $\beta \;and\;\beta^{2}$

Figure 7a,b shows the results of β and $\beta^{2}$ determination, respectively, before and after the corrections for diffraction and attenuation are performed. Here, β and $\beta^{2}$ were measured from the amplitudes of the second and third harmonic waves, respectively, using the same output signal. The effect of source nonlinearity correction is not considered here and is discussed separately in the next section. The uncorrected β shows a decreasing and then increasing behavior with increasing sample thickness. It can be observed that the attenuation and diffraction corrections shift large and small values of β that deviate from the mean closer to the mean value of 6.03. The mean value of β after the corrections is found to be 5.76. This behavior agrees well with the previous β measurement results [19].

The effect of the diffraction and attenuation corrections on the cubic nonlinearity parameter $\beta^{2}$ is similarly shown in Figure 7b. The uncorrected $\beta^{2}$ shows relatively uniform values between 6 cm and 12 cm in sample thickness and increases suddenly from these values at sample thicknesses shorter than 6 cm. The reason for this large deviation at a short distance is basically the source nonlinearity included in the third harmonic and the generation of an insufficient third harmonic component from the specimen due to the short propagation distance. This problem can be improved to some extent by removing the source nonlinearity in the $\beta^{2}$ calculation, as is demonstrated later. The influence of the diffraction and attenuation corrections is basically small in the uniform $\beta^{2}$ region and tends to increase as the sample thickness becomes shorter. The mean value of the corrected $\beta^{2}$ in the 6 cm to 12 cm region is 40.66, which is approximately 1.2 times larger than the square of the corrected β.

The uncorrected and corrected $\beta^{2}$ results start to show a gradual and then sudden increasing behavior at the sample thickness of 6 cm and shorter, as shown in Figure 7b. This is due to the noise floor of the measurement system and the low third harmonic generation at short sample thicknesses. However, in the case of β measured from the second-harmonic generation, this trend did not occur due to the second harmonic amplitudes being generated greater than the noise floor even for the shortest 2 cm thick sample, as shown in Figure 7a. Through repeated measurements, we were able to measure β with less than 5% uncertainty and $\beta^{2}$ with less than 10% uncertainty. The error bars were not marked in the accompanying figures here.

The $\beta^{2}$ measurement from the third harmonic amplitude is more challenging in many ways than the β measurement from the second harmonic amplitude due to the need for a more accurate measurement of the acoustic parameters, such as sample thickness, fundamental wave displacement, and acoustic velocity, because these variables are squared, cubed, and quadrupled in the $\beta^{2}$ formula. The same phenomenon occurs in the diffraction and attenuation corrections for the fundamental wave. Therefore, the measurement accuracy of these parameters generally has a more serious effect on the measured value of $\beta^{2}$. Considering the high sensitivity of experimental variables, an error of the $\beta^{2}$ measurement that is approximately twice as large as that of the β measurement can be acceptable.

### 4.6. Comparison of Cubic Nonlinearity Paramter $\beta^{2}$ and Square of Quadratic Nonlinearity Parameter β∗ β

It is interesting to compare the directly measured $\beta^{2}$ using the third harmonic amplitude with the square of β, β∗β, measured from the second harmonic amplitude. The initial and corrected results of these parameters are presented in Figure 8a,b, respectively, for comparison. It can be observed in Figure 8a that the initial results from two sets of measurement show a similar behavior for sample thicknesses larger than 6 cm. The largest difference in this region is approximately 20.4%, occurring at a 6 cm sample thickness, and the difference becomes smaller as the sample thickness increases. The $\beta^{2}$ and β∗β results after being corrected for diffraction and attenuation are presented in Figure 8b. Since the effect of the diffraction and attenuation corrections on $\beta^{2}$ is basically small, the agreement between these two values does not improve and remains almost the same. Another major reason is the significant amount of source nonlinearity contained in the amplitude of the third harmonic. We will show later a much better agreement between $\beta^{2}$ and β∗β over a wider range of sample thicknesses and input power levels by removing the source nonlinearity.

### 4.7. Source Nonlinearity Check and Correction

When the displacement amplitudes are available at the fundamental and second harmonic frequencies, an appropriate check for source nonlinearity is to plot the second harmonic amplitude ($U_{2}$) as a function of the square of the fundamental amplitude ($U_{1}^{2}$) at the different input levels used in the measurement. This kind of plot provides insight into the system and sample response. At the higher amplitudes, the response should be linear if the sample behaves as a classically nonlinear solid. When this plot is linearly extrapolated, the y-intercept indicates the noise floor of the measurement system and/or the source nonlinearity involved in the measurement of the second harmonic amplitude [19]. Similarly, the third harmonic amplitude ($U_{3}$) as a function of the cube of the fundamental amplitude ($U_{1}^{3}$) at different input levels can be plotted to check the existence of the noise floor and/or source nonlinearity included in the measurement of the third harmonic amplitude.

Figure 9a,b shows the typical plot of $U_{1}^{2}$ vs. $U_{2}$ and $U_{1}^{3}$ vs. $U_{3}$, respectively, for the 8 cm sample at the nine input power levels used. The best-fit straight line is also shown in each figure, and there exists a good linearity between these data. Since the y-intercept almost passes through the origin in Figure 9a, the source nonlinearity, which might be contained in the second harmonic amplitude, can be ignored. This means that the measurement system used here exhibits a relatively low noise floor, and a very small amount of extraneous second harmonic was produced. However, the y-intercept in Figure 9b is well above the origin, indicating that a significant amount of source nonlinearity is contained in the third harmonic amplitude. The cause of this source nonlinearity can be explained as follows. When a finite amplitude narrowband toneburst with a fundamental frequency *f*_0_ is applied to the transmitter, a noticeable 3*f*_0_ component is generated due to the odd-numbered resonance of the transmitter. This also propagates as the linear wave of frequency 3*f*_0_. This wave will be added up to the nonlinear third harmonic wave generated in the solid specimen, sometimes giving an excessively large third harmonic amplitude or cubic nonlinearity parameter.

A plot, as shown in Figure 9, was prepared for each specimen, and the y-intercept was obtained by fitting the curve with a straight line to confirm the existence of source nonlinearity for the second and third harmonic amplitudes. If a source nonlinearity is found to exist, this source nonlinearity should be subtracted from the corresponding harmonic amplitude. Then, diffraction and attenuation corrections are performed according to Equations (12) and (13) to obtain all corrected $\beta \;\mathrm{and}\;\beta^{2}$.

Since the effects of the diffraction and attenuation corrections on these parameters were discussed in the previous section, only the effect of the source nonlinearity correction will be discussed here. No source nonlinearity correction will be performed for the second harmonic amplitude because the source nonlinearity was not included in $U_{2}$, as shown in Figure 9a. Therefore, the value of the corrected β will be the same as the value obtained earlier in Figure 7a and Figure 8a. The effect of the source nonlinearity correction for $U_{3}$ on the cubic nonlinearity parameter $\beta^{2}$ and its relationship with the square of β of the quadratic nonlinearity parameter will be examined.

The corrections applied here are diffraction, attenuation, and source nonlinearity corrections. Only the diffraction and attenuation corrections were performed for β, while all three corrections were performed for $\beta^{2}$. First, Figure 10a compares $\beta^{2}$ with three different corrections as a function of sample thickness—no corrections, diffraction (D), and attenuation (M) corrections, and all corrections including the source nonlinearity correction.

The diffraction and attenuation corrections shift the initial uncorrected value of $\beta^{2}$ to slightly lower or slightly higher values depending on the sample thickness, but the amount of corrections are very small, especially in the 6 cm to 12 cm region. In contrast, the addition of source nonlinearity correction lowers the value of $\beta^{2}$ much more over all sample thicknesses, and the amount of reduction increases as the sample thickness becomes thinner. Due to the source nonlinearity correction, the cubic nonlinearity parameter $\beta^{2}$ now agrees better with the square of the quadratic nonlinearity parameter β∗β down to the 4 cm thickness.

With the help of source nonlinearity correction, the value of $\beta^{2}$ now extends to be valid down to the 4 cm thickness, and the approximate relationship $\beta^{2}\approx \beta *\beta$ appears to be maintained in the interval where $\beta^{2}$ and β∗β are both valid. Based on the measurement data for AL 6061 covered in this study, these two parameters differ by approximately 8.05% when calculated using the average values in this effective interval. The average $\beta^{2}\;$and β∗β of the five samples ranging from 4 cm to 12 cm thickness are 30.33 and 32.77, respectively. On the other hand, if the mean value of $\sqrt{\beta^{2}}$ is compared with the mean value of the directly measured β, the difference is less than 4%.

Figure 10b shows the behavior of all corrected $\beta^{2}$ as a function of the input power level and compares with the square of the corrected β, β∗β. These results were obtained from the measurement on the 4 cm thick sample. Other samples are expected to exhibit similar behavior. The effect of the diffraction and attenuation corrections is small, and the uncorrected and corrected $\beta^{2}$ have extremely large values and are getting worse at the lower power levels compared to the corrected β∗β.

The source nonlinearity correction greatly reduces both the uncorrected and corrected $\beta^{2}$ values, and this correction effect becomes much more evident as the power level goes lower. Now, $\beta^{2}$ shows relatively uniform values down to a power level of approximately 15 and agrees well with the value of the square of the quadratic nonlinearity parameter β∗β.

The measurement results of $\beta^{2}$ shown in Figure 10 clearly demonstrate that confirming the existence of source nonlinearity in the measured third harmonic amplitude and properly correcting it are very important for the accurate and reliable analysis of third harmonic generation measurement results.

In nonlinear ultrasonic measurement, it is basically necessary to check the presence of source nonlinearity in the received output signal. In particular, when a piezoelectric transducer is used to generate odd-numbered harmonics of a fundamental wave such as the 3rd order it is essential to check and remove the source nonlinearity since some degree of source nonlinearity cannot be avoided. The source nonlinearity suppression or elimination methods currently proposed in nonlinear ultrasound measurement include the acoustic modeling-based approach, the harmonic cancellation method, and metamaterial-based bandgap filtering. Jeong et al. [30] and Song et al. [31] were able to extract the source nonlinearity included in the measured second harmonic amplitude by comparing the simulated nonlinear longitudinal wave fields with the experimental data. Torello et al. [32] used a similar acoustic modeling approach to remove the source nonlinearity contained in the measured nonlinear surface wave fields. Tang and Clement [33] reported a harmonic cancellation technique by using a switched-mode power converter without an additional output filter. More recently, the metamaterial-based acoustic filter was designed and applied in the form of superlattices [34], metasurface [35], waveguide rods [36], and additively manufactured phononic materials [37]. These methods require modeling of nonlinear ultrasound beam fields and additional hardware to the experimental setup. Compared to these methods, the source nonlinearity correction method used in this study is convenient to apply and has a clear advantage because it is performed through a little processing of experimental data measured in the input voltage range used for harmonic generation.

## 5. Conclusions

In this work, the effect of source nonlinearity corrections on the measurement of the cubic nonlinearity parameter $\beta^{2}$ is presented for aluminum specimens of various thicknesses at various input power levels. By correcting the source nonlinearity contained in the third harmonic amplitude and further verifying the approximate relationship of $\beta^{2}\approx \beta *\beta$, it was shown that the cubic nonlinearity parameter could be reliably determined even at thinner samples and lower input voltages. Based on our current nonlinearity measurement results, we believe that, if the cubic nonlinearity parameter $\beta^{2}$ is measured correctly together with the source nonlinearity correction, we can obtain the quadratic nonlinearity parameter β from the measured $\beta^{2}$ within a 5–10% difference.

When a piezoelectric transmitter is used in a nonlinear ultrasonic measurement system, the occurrence of source nonlinearity that is associated with third harmonic generation appears to be unavoidable. Therefore, it is essential to eliminate the source nonlinearity in the calculation of cubic nonlinearity parameters. We proposed a method for confirming the existence of source nonlinearity in the measured third harmonic amplitude and correcting it. Compared to the existing methods for source nonlinearity suppression or elimination, the source nonlinearity correction method proposed in this study is convenient to apply and has a clear advantage because it is performed through a little processing of experimental data measured in the input voltage range used for harmonic generation.

## Figures and Tables

**Figure 1 materials-16-04453-f001:**
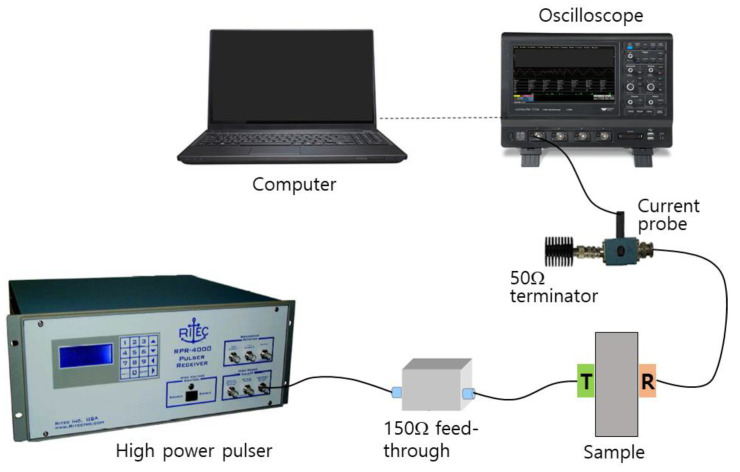
Block diagram of the experimental setup for harmonic generation measurement using a finite amplitude, through-transmission method.

**Figure 2 materials-16-04453-f002:**
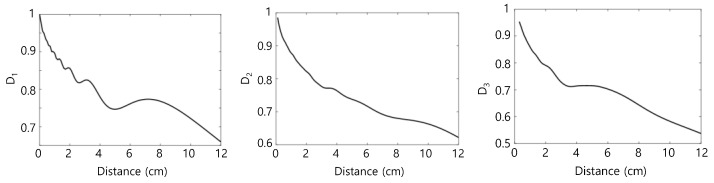
Calculated diffraction corrections for fundamental (*D*_1_), second harmonic (*D*_2_), and third harmonic (*D*_3_) waves: Frequency = 5 MHz, transmitter and receiver diameters = 0.5 inch, material = Al 6061 (L-wave velocity= 6450 m/s).

**Figure 3 materials-16-04453-f003:**
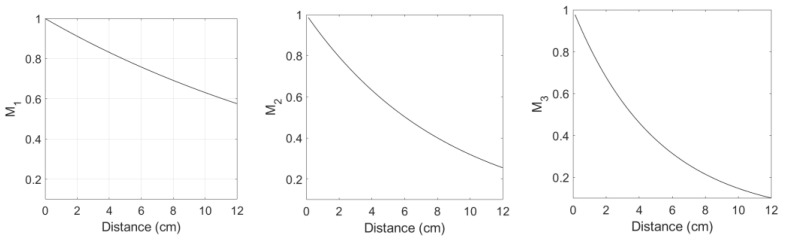
Calculated attenuation corrections for fundamental (*M*_1_), second harmonic (*M*_2_), and third harmonic (*M*_3_) waves when the fundamental frequency is 5 MHz.

**Figure 4 materials-16-04453-f004:**
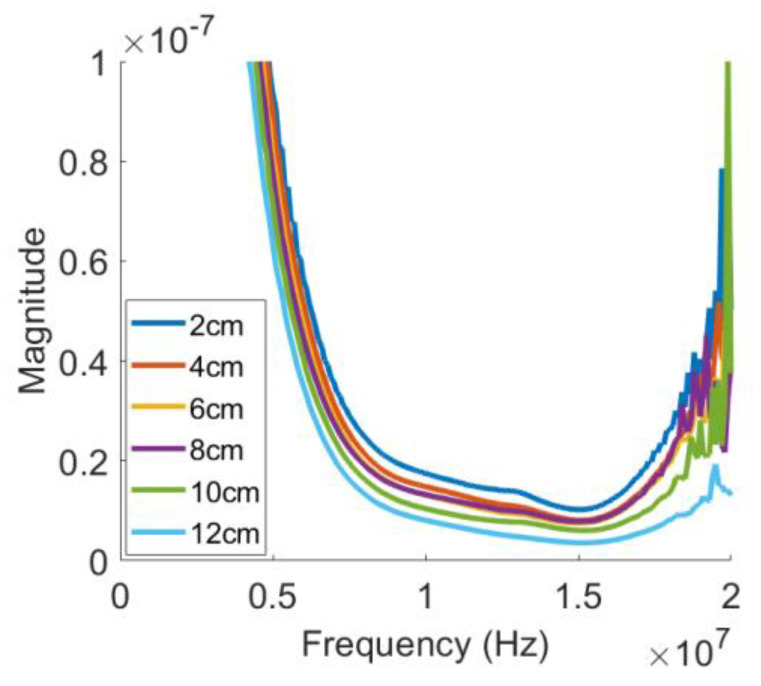
Receiver transfer functions obtained for all six specimens. Diffraction and attenuation effects were appropriately corrected.

**Figure 5 materials-16-04453-f005:**
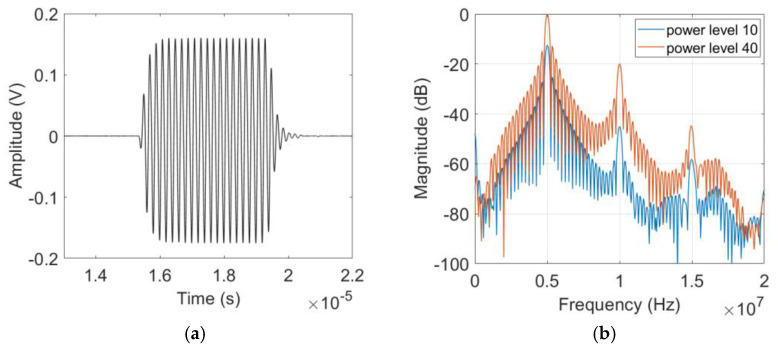
Typical results of (**a**) Current output signal and (**b**) Magnitude spectrum at two different input power levels measured on the 8 cm sample.

**Figure 6 materials-16-04453-f006:**
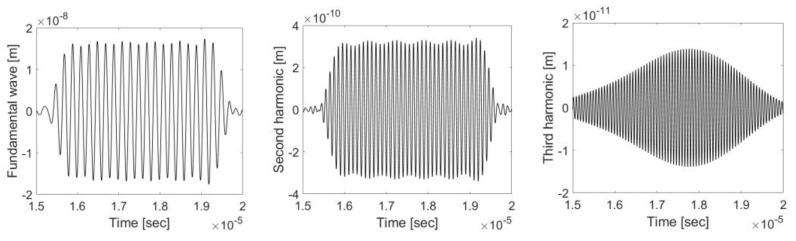
Extracted fundamental, second harmonic, and third harmonic displacement waveforms.

**Figure 7 materials-16-04453-f007:**
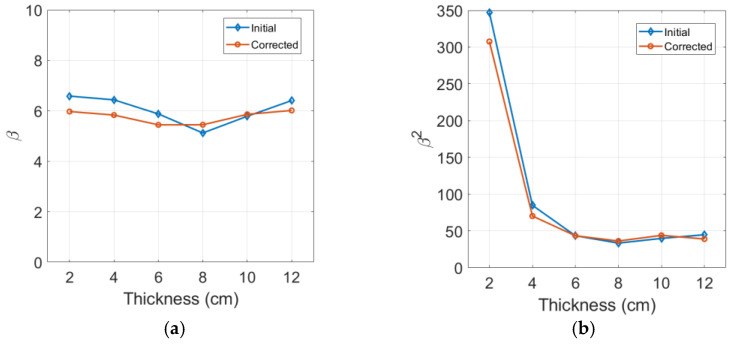
Measured nonlinearity parameters and effects of diffraction and attenuation corrections on (**a**) β results and (**b**) $\beta^{2}$ results.

**Figure 8 materials-16-04453-f008:**
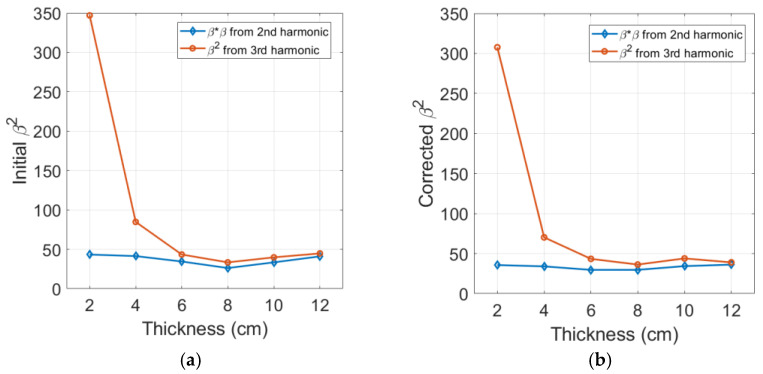
Comparison of the directly measured $\beta^{2}$ with the square of the directly measured β: (**a**) Before and (**b**) After corrections for diffraction and attenuation.

**Figure 9 materials-16-04453-f009:**
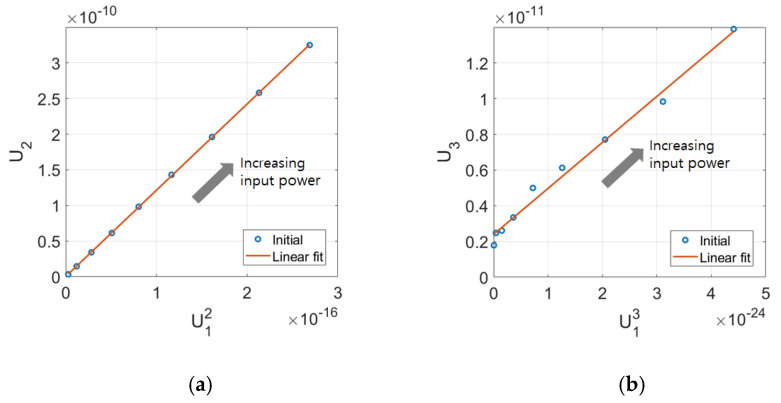
Typical examples of the source nonlinearity check on the 8 cm sample: (**a**) Plot of $U_{2}$ as a function of $U_{1}^{2}$ and (**b**) Plot of $U_{3}$ as a function of $U_{1}^{3}$ at the various input power levels used. The best-fit straight line is also shown in each figure.

**Figure 10 materials-16-04453-f010:**
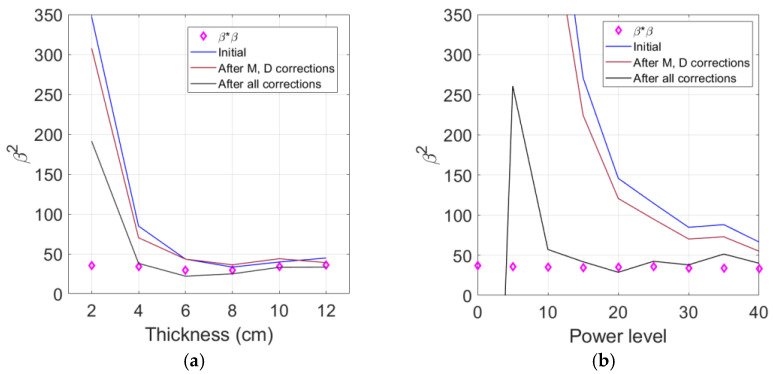
Comparison of all corrected $\beta^{2}$ with corrected β∗β: As a function of (**a**) sample thickness and (**b**) input power level.

## Data Availability

The data presented in this study are available on request from the corresponding author. The data are not publicly available due to privacy.

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
