# Peer review of "Measurement and In-Depth Analysis of Higher Harmonic Generation in Aluminum Alloys with Consideration of Source Nonlinearity"

_materials, 2023, doi:10.3390/ma16124453_

Round 1
Reviewer 1 Report
Nondestructive evaluation method using ultrasound technique was investigated by focusing on the nonlinearity in higher harmonic generation. The relationship between the nonlinearity and the certain material state or the assumed micro-damage should be discussed to clearly insist the utility values of investigated results.
The following comments were found ;
1) The accuracy of measured values is likely to be dependent on the contacting condition of the sensors on the specimens, including the interface condition. It is better to describe how much the authors took account of them. In particular, in order to discuss on the nonlinearity on the measured values, if even referring some papers, they should mention repeatedly because of the critical importance.
2) The nonlinearity is very dependent on the intensity of input source. Describe the dependency on the intensity on the nonlinearity characteristics.
3) As for line 419, it is better to explain the concerned mechanism of the source nonlinearities.
Author Response
Authors’ Reply to Reviewer 1
The authors of this paper are very grateful for the valuable comments from the reviewer. We took the review results very carefully and used our best knowledge to answer all the comments and questions. The paper has been revised as much as possible to reflect the reviewer's opinion.
The revised part of the paper is marked with blue letters in the manuscript.
Nondestructive evaluation method using ultrasound technique was investigated by focusing on the nonlinearity in higher harmonic generation. The relationship between the nonlinearity and the certain material state or the assumed micro-damage should be discussed to clearly insist the utility values of investigated results.
The following comments were found:
1) The accuracy of measured values is likely to be dependent on the contacting condition of the sensors on the specimens, including the interface condition. It is better to describe how much the authors took account of them. In particular, in order to discuss on the nonlinearity on the measured values, if even referring some papers, they should mention repeatedly because of the critical importance.
Yes, we agree. The measured values depends on the contact condition between the transducers and the specimen. In connection with this, the following paragraphs were added in new Section 3.6.
3.6. Comments on contact nonlinear ultrasound testing
“ In contact nonlinear ultrasound testing, the contact and interface conditions between the specimen and the transducer can have a significant impact on the measured harmonic amplitudes. The contact and interface conditions may include the surface roughness of the specimen [28], the type/amount/contact holding time of couplant [29], and the intensity of the contact pressure. In the case of transmission measurement, it is necessary to consider both the transmission side and the reception side. When measuring absolute nonlinearity parameters, it is necessary to ensure that the calibration measurement state of the receiver remains the same during harmonic generation measurements. The first author of this paper has conducted research on nonlinear ultrasound tests for many years and has established a nonlinearity parameter measurement technology with excellent accuracy and repeatability. Some of them were used in this study as described below.
In order to minimize the effect of surface roughness, it is necessary to maintain the same surface roughness on each specimen as much as possible. The prepared specimens were machined so that the upper and lower surfaces were parallel. The surface roughness of each specimen was maintained at the same level as possible using a metal abrasive. A thin layer of couplant is applied to the transducer surface, and the transducer is maintained in a pressurized state with a constant pressure using a pressurization device. Experimental data are acquired after the pressed couplant reaches a steady state. This time usually takes several minutes. A specially designed pressurization fixture is used so that the receiver and transmitter are pressed separately. The pressurization state of the receiver during the calibration measurement remains the same throughout the harmonic generation measurement. A pin spring type fixing and pressurization device is devised and used for pressurization of piezoelectric elements such as bare crystals in the transmission side. Using this set of contact and boundary conditions, the quadratic nonlinearity parameter can be measured with less than 5% uncertainty, while the cubic nonlinearity parameter can be measured with less than 10% uncertainty.”
2) The nonlinearity is very dependent on the intensity of input source. Describe the dependency on the intensity on the nonlinearity characteristics.
The dependence of nonlinearity or the received harmonic amplitudes on the intensity of input source can be seen in Figs. 9(a) and 9(b), where and are plotted at nine input power levels used. The fundamental amplitude is directly proportional to the input source, so the values of and become larger as the input power level increases.
“Increasing input power” was added in Figs. 9(a) and 9(b).
3) As for line 419, it is better to explain the concerned mechanism of the source nonlinearities.
The mechanism of the source nonlinearity is explained in Section 4.7, line number 482 to 488.
“The cause of this source nonlinearity can be explained as follows. When a finite amplitude narrowband toneburst with a fundamental frequency f0 is applied to the transmitter, a noticeable 3f0 component is generated due to the odd-numbered resonance of the transmitter. This also propagates as the linear wave of frequency 3f0. This wave will be added up to the nonlinear third harmonic wave generated in the solid specimen, giving sometimes an excessively large third harmonic amplitude or cubic nonlinearity parameter.”

Reviewer 2 Report
I have the following questions:
In ultrasonic testing, frequencies in the range of 20-100 kHz are usually used. Why was 5 MHz used as the base frequency?
The tests presented in the article were carried out on aluminum samples, were there or are there any tests planned for other materials, e.g. composites? What frequencies will then be used?
Author Response
Authors’ Reply to Reviewer 2
The authors of this paper are very grateful for the valuable comments from the reviewer. We took the review results very carefully and used our best knowledge to answer all the comments and questions. The paper has been revised as much as possible to reflect the reviewer's opinion.
The revised part of the paper is marked with blue letters in the manuscript.
I have the following questions:
In ultrasonic testing, frequencies in the range of 20-100 kHz are usually used. Why was 5 MHz used as the base frequency?
The tests presented in the article were carried out on aluminum samples, were there or are there any tests planned for other materials, e.g. composites? What frequencies will then be used?
It is known that ultrasonic waves in the kHz range are mainly used for nonlinear experiments using Lamb waves. Fundamental waves in the MHz range are commonly used in nonlinear experiments using longitudinal waves. Since the magnitude of harmonics generated in a nonlinear experiment is proportional to the frequency, the more high frequencies are used, the more advantageous it is to generate nonlinear ultrasound. Another reason is to identify multiple reflected waves when a tone burst waveform of several tens of cycles is used in a transmission or pulse-echo nonlinear longitudinal wave experiment. In addition, since the magnitude of reflected or scattered waves generated by the interaction of ultrasonic waves with defects such as cracks is proportional to the frequency of the incident wave, it is advantageous to use high frequencies. Ultrasound in the MHz range is also used in nonlinear surface wave experiments.
We do not plan to apply nonlinear ultrasound to composite materials for the time being. Since composites are mainly in the form of laminates, Lamb waves are more suitable for nonlinear ultrasonic testing.

Reviewer 3 Report
Comments to the authors:
1. Group citation must be avoided Ex. [13-18]
2. All the equations need to be cited properly.
3. What is the transfer function obtained from Receiver Calibration [??(?)]
4. What is the order and type of the transfer function?
5. Figure 1 quality needs to be improved.
6. In line no 203, Fig 3 has to be changed to Figure 1.
7. All transfer functions and their significance need to be mentioned.
8. What about the stability of the system, when the order increases?
9. How the authors addressed the source uncertainty?
10. A detailed discussion section is mandatory.
11. Concluison has to be supported with obtained results.
Moderate editing of the English language is required.
Author Response
Authors’ Reply to Reviewer 3
The authors of this paper are very grateful for the valuable comments from the reviewers. We took the review results very carefully and used our best knowledge to answer all the comments and questions. The paper has been revised as much as possible to reflect the reviewer's opinion.
The revised part of the paper is marked with blue letters in the manuscript.
- Group citation must be avoided Ex. [13-18]
References [13-18] were subdivided as follows:
“In addition to longitudinal waves, third harmonic generation using Lamb waves and Rayleigh surface waves was studied for different application purposes. Lissenden et al. [13] studied the effect of microstructure evolution on higher harmonic generation of guided waves. Zhao et al. [14] used third harmonic Lamb waves for early fatigue damage detection. Li et al. [15] and Wen et al. [16] applied third harmonic shear horizontal waves for material degradation monitoring. Achenbach and Wang [17] and Wang and Achenbach [18] used third harmonic surface waves for the characterization of incompressible material of cubic nonlinearity and interface conditions, respectively.”
- All the equations need to be cited properly.
References were added to equations.
Equations (3)-(5) have been derived ourselves.
- What is the transfer function obtained from Receiver Calibration [?(?)]
The purpose of the transfer function from the receiver calibration measurement is well described in Section 3.1.
- What is the order and type of the transfer function?
There is no specific transfer function for a specific order. The measured transfer function can be used to calculate all orders of harmonic waves.
- Figure 1 quality needs to be improved.
Figure 1 was replaced with new one.
- In line no 203, Fig 3 has to be changed to Figure 1.
Figure 3 was changed to Figure 1.
- All transfer functions and their significance need to be mentioned.
As answered in Q3 above, the purpose and significance of transfer function are well described in Section 3.1 and Section 4.2.
- What about the stability of the system, when the order increases?
Since all harmonic waves are measured using one receiver in one experimental setup, the stability of the system according to the order is not affected. However, in the calculation of nonlinearity parameter values according to the order, for example, in the case of quadratic and cubic nonlinearity parameter calculations, the cubic nonlinearity parameter is affected much more by the measured values such as wave number k, specimen thickness x, fundamental wave displacement U1, diffraction correction D3, etc.
- How the authors addressed the source uncertainty?
The cause of source nonlinearity is explained in Section 3.5. The method of checking whether the source nonlinearity is included in the measured harmonic amplitudes and the method of correcting (removing) it are explained in detail in Section 4.7 and figure 9.
- A detailed discussion section is mandatory.
Analytical and experimental results were provided in Sections 4.1 through 4.7, and detailed discussions were also accompanied. In addition, the following discussion were added in the revised manuscript.
“In nonlinear ultrasonic measurement, it is basically necessary to check the presence of source nonlinearity in the received output signal. In particular, when a piezoelectric transducer is used to generate odd-numbered harmonics of a fundamental wave, such as the 3rd order, it is essential to check and remove the source nonlinearity since some degree of source nonlinearity cannot be avoided. Source nonlinearity suppression or elimination methods currently proposed in nonlinear ultrasound measurement include acoustic modeling-based approach [30-32], harmonic cancellation method [33], and metamaterial-based bandgap filtering [34-37]. These methods require a modeling of nonlinear ultrasound beam fields and additional hardware to the experimental setup. Compared to these methods, the source nonlinearity correction method used in this study is convenient to apply and has a clear advantage because it is performed through a little processing of experimental data measured in the input voltage range used for harmonic generation.”
- Conclusion has to be supported with obtained results.
The second paragraph of the conclusion was modified as follows:
“When a piezoelectric transmitter is used in a nonlinear ultrasonic measurement system, the occurrence of source nonlinearity associated with third harmonic generation appears to be unavoidable. Therefore, it is essential to eliminate the source nonlinearity in the calculation of cubic nonlinearity parameters. We proposed a method for confirming the existence of source nonlinearity in the measured third harmonic amplitude and correcting it. Compared to the existing methods for source nonlinearity suppression or elimination, the source nonlinearity correction method proposed in this study is convenient to apply and has a clear advantage because it is performed through a little processing of experimental data measured in the input voltage range used for harmonic generation.”

Reviewer 4 Report
In this paper, the effect of the source nonlinearity approach on ductile polycrystalline materials includes receiver calibration, diffraction, attenuation correction, and more importantly source nonlinearity correction. The nonlinearity sources are divided into three origins, the source base, the medium geometry, and material behavior. The current manuscript focused on source nonlinearity. After the following minor comments, I agree to publish in the journal. 1. The work presented by the authors is good. I didn't notice any major technical flaws. But few language errors (Grammar, prepositions, and articles) were observed. Hence the paper should be thoroughly checked and corrected. 2. In section 4.1, reference 21 pointed to the variation of diffraction correction as a function of propagation distance. As a reader, I expect authors to present in one short sentence, what is "diffraction correction"? I recommend adding an explanation. Of course, for detail, I will refer to reference 21. 3. In Figure 4, various samples with different thicknesses were utilized. It is recommended to explain the effect of thickness on the results. 4. As can be seen in the title, "ductile polycrystalline materials" was used. Would you please explain why in the body of manuscripts discussed only aluminum 6061 T6? Is it not better to replace ductile polycrystalline materials with aluminum 6061 T6? The authors need to clarify. 5. An outline of the paper at the end of the introduction section is recommended.
Author Response
Authors’ Reply to Reviewer 4
The authors of this paper are very grateful for the valuable comments from the reviewers. We took the review results very carefully and used our best knowledge to answer all the comments and questions. The paper has been revised as much as possible to reflect the reviewer's opinion.
The revised part of the paper is marked with blue letters in the manuscript.
In this paper, the effect of the source nonlinearity approach on ductile polycrystalline materials includes receiver calibration, diffraction, attenuation correction, and more importantly source nonlinearity correction. The nonlinearity sources are divided into three origins, the source base, the medium geometry, and material behavior. The current manuscript focused on source nonlinearity. After the following minor comments, I agree to publish in the journal.
- The work presented by the authors is good. I didn't notice any major technical flaws. But few language errors (Grammar, prepositions, and articles) were observed. Hence the paper should be thoroughly checked and corrected.
The paper was thoroughly checked for grammar, prepositions and articles, and corrected as best as we can.
- In section 4.1, reference 21 pointed to the variation of diffraction correction as a function of propagation distance. As a reader, I expect authors to present in one short sentence, what is "diffraction correction"? I recommend adding an explanation. Of course, for detail, I will refer to reference 21.
The diffraction correction was defined in one short sentence. Reference 21 was cited to point out the variation of diffraction correction as a function of propagation distance.
“Diffraction correction is defined as the amplitude of the actual wave divided by that of the plane wave with both wave amplitudes received at the same propagation distance in a nonattenuating medium. Diffraction effects generally depend on the size of the transmitter and receiver, frequency and propagation distance. An exact integral expression exists for the linear field when both transmitter and receiver sizes are the same [26, 27]. The diffraction corrections for both fundamental and higher harmonic waves have been developed and can be efficiently used in a wide range of transmitter-receiver geometries [21]. The variation of diffraction correction as a function of propagation distance is shown in [21].”
- In Figure 4, various samples with different thicknesses were utilized. It is recommended to explain the effect of thickness on the results.
The effect of sample thickness on the results was explained as follows:
“Since the magnitude spectrum is also given as a function of plane wave term, exp(ikz), where z is the propagation distance, it shows a dependence on the sample thickness or propagation distance. At a given frequency, the spectral values become smaller as the sample thickness increases.”
- As can be seen in the title, "ductile polycrystalline materials" was used. Would you please explain why in the body of manuscripts discussed only aluminum 6061 T6? Is it not better to replace ductile polycrystalline materials with aluminum 6061 T6? The authors need to clarify.
The title of the paper was modified as follows:
“Measurement and in-depth analysis of higher harmonic generation in aluminum alloys with consideration of source nonlinearity”
- An outline of the paper at the end of the introduction section is recommended.
An outline of the paper was added at the end of the introduction section:
“The rest of this paper is organized as follows. Section 2 briefly introduces the plane wave solutions for nonlinear wave equation and the definition of quadratic (β) and cubic () nonlinearity parameters with corrections for attenuation and diffraction. Section 3 describes the elements of absolute measurement of nonlinearity parameters: Receiver calibration measurement; harmonic generation measurement; absolute displacement measurement; diffraction and attenuation corrections; check for source nonlinearity and correction. Section 4 presents the in-depth analysis of experimental results with emphasis on the correction of source nonlinearity for accurate comparison of and β*β. Lastly, conclusions are outlined in Section 5. “

Round 2
Reviewer 3 Report
Congrats to the authors.
I suggest the authors replace the group citation [2-12], [30-32], and [34-37] and brief it separately.
No specific comments.
Author Response
Authors’ Reply to Reviewer 3 (2)
The authors of this paper are very grateful for the valuable comments from the reviewers. We took the review results very carefully and used our best knowledge to answer all the comments and questions. The paper has been revised as much as possible to reflect the reviewer's opinion.
The revised part of the paper is marked with blue letters in the manuscript.
I suggest the authors replace the group citation [2-12], [30-32], and [34-37] and brief it separately.
The references were subdivided as follows:
The advantages of using the cubic nonlinearity parameter for fatigue cracks [2,3], plastic deformation [4-6], microstructures [7-9], dislocation [10], and precipitation [11,12] of metals were discussed by comparing the relevant results of the quadratic nonlinearity parameter.
Jeong et al. [30] and Song et al. [31] were able to extract the source nonlinearity included in the measured second harmonic amplitude by comparing the simulated nonlinear longitudinal wave fields with the experimental data. Torello et al. [32] used a similar acoustic modeling approach to remove the source nonlinearity contained in the measured nonlinear surface wave fields. Tang and Clement [33] reported a harmonic cancellation tech-nique by using a switched-mode power converter without an additional output filter. More recently, the metamaterial-based acoustic filter was designed and applied in the form of superlattices [34], metasurface [35], waveguide rods [36], and additively manufactured phononic materials [37].
